# Methylated Cell-Free DNA Sequencing (MeD-seq) of LpnPI Digested Fragments to Identify Early Progression in Metastatic Renal Cell Carcinoma Patients on Watchful Waiting

**DOI:** 10.3390/cancers15051374

**Published:** 2023-02-21

**Authors:** Manouk K. Bos, Sarah R. Verhoeff, Sjoukje F. Oosting, Willemien C. Menke-van der Houven van Oordt, Ruben G. Boers, Joachim B. Boers, Joost Gribnau, John W. M. Martens, Stefan Sleijfer, Carla M. L. van Herpen, Saskia M. Wilting

**Affiliations:** 1Department of Medical Oncology, Erasmus MC Cancer Institute, Erasmus University Medical Center, 3015 GD Rotterdam, The Netherlands; 2Department of Medical Oncology, Radboud University Medical Center, 6525 GA Nijmegen, The Netherlands; 3Department of Medical Oncology, University Medical Centre Groningen, 9713 GZ Groningen, The Netherlands; 4Department of Medical Oncology, Cancer Center Amsterdam, Amsterdam UMC, Vrije Universiteit Amsterdam, 1081 HZ Amsterdam, The Netherlands; 5Department of Developmental Biology, Erasmus Medical Center, 3015 GD Rotterdam, The Netherlands

**Keywords:** metastatic renal cell carcinoma, watchful waiting, DNA methylation, cfDNA

## Abstract

**Simple Summary:**

A subset of patients with an intermediate risk for renal cell carcinoma have an indolent disease course. For these patients, the initiation of treatment for metastatic disease can be safely delayed by entering a watchful waiting period. At present, we are not able to identify those patients with indolent disease and the minor group of patients with rapidly progressive disease requiring earlier initiation of systemic treatment. In this study, we investigate whether cell-free DNA (cfDNA) can be used as a blood-based biomarker to identify those patients with rapid progression. Methylated cfDNA profiles were used as a proxy for tumor load in blood. cfDNA methylation patterns were associated with the time to radiological progression, but not with the watchful waiting time. The results of this study do not provide definite proof that cfDNA methylation patterns are associated with WW time.

**Abstract:**

According to the current guidelines, watchful waiting (WW) is a feasible option for patients with good or intermediate prognosis renal-cell carcinoma (RCC). However, some patients rapidly progress during WW, requiring the initiation of treatment. Here, we explore whether we can identify those patients using circulating cell-free DNA (cfDNA) methylation. We first defined a panel of RCC-specific circulating methylation markers by intersecting differentially methylated regions from a publicly available dataset with known RCC methylation markers from the literature. The resulting RCC-specific methylation marker panel of 22 markers was subsequently evaluated for an association with rapid progression by methylated DNA sequencing (MeD-seq) in serum from 10 HBDs and 34 RCC patients with a good or intermediate prognosis starting WW in the IMPACT-RCC study. Patients with an elevated RCC-specific methylation score compared to HBDs had a shorter progression-free survival (PFS, *p* = 0.018), but not a shorter WW-time (*p* = 0.15). Cox proportional hazards regression showed that only the International Metastatic Renal Cell Carcinoma Database Consortium (IMDC) criteria were significantly associated with WW time (HR 2.01, *p* = 0.01), whereas only our RCC-specific methylation score (HR 4.45, *p* = 0.02) was significantly associated with PFS. The results of this study suggest that cfDNA methylation is predictive of PFS but not WW.

## 1. Introduction

Systemic treatment options for patients with metastatic renal clear cell carcinoma (mccRCC) have changed within the last two decades, with the arrival of angiogenesis inhibitors and immunotherapy [1,2]. Those treatments can be given either sequentially as monotherapy or as combination therapy, depending on the patients’ risk status, as assessed by the International Metastatic Renal Cell Carcinoma Database Consortium (IMDC) criteria [3,4]. However, the immediate initiation of systemic treatment at the diagnosis of metastatic disease is not always indicated. A selected subset of mccRCC patients have an indolent disease course that can last for years. As such, this subgroup of patients can benefit from active surveillance, also referred to as watchful waiting (WW). A period of WW postpones the potentially unnecessary treatment toxicity and is considered an initial treatment strategy in newly diagnosed patients with mRCC, according to the current clinical practice guidelines [5].

In daily practice, it is challenging to identify patients with expected indolent disease and potential candidates for the WW strategy. It is clear that patients with a poor prognosis, according to the IMDC score, require immediate systemic treatment. However, the remaining patients with a good or intermediate prognosis based on the IMDC score form a heterogeneous population [4]. This includes patients with more aggressive disease who progress within a few months after diagnosis, and patients with an indolent course of disease over several years. This indicates that the IMDC criteria alone do not suffice in selecting patients for long-term WW [6].

While only limited prospective studies have studied WW in mccRCC patients, one prospective study has suggested that disease extent (number of involved organs) could be a valuable prognostic factor to discriminate between indolent and more aggressive disease [6]. Recently, the data of the prospective Imaging Patients for Cancer drug Selection (IMPACT)-RCC study confirmed the predictive value of <2 IMDC risk factors and ≤2 involved organ sites (or “W&W criteria”) to select the patients who are eligible for a period of WW [7]. In addition to these criteria, this study showed the potential of tumor [18F]FDG-uptake to predict the WW time beyond these “W&W criteria”.

In addition to imaging, cell-free DNA (cfDNA) is a biomarker that could be used to study tumor characteristics in a minimally invasive way. CfDNA is composed of DNA fragments originating from dying cells in body fluids, such as blood and urine [8]. With respect to blood, cfDNA can be obtained from either serum or plasma, with a preference for the latter when available, given the higher background of leukocyte DNA from serum [9]. Given the low fractions at which tumor-derived cfDNA is present in the circulation, especially in patients with RCC [10], highly sensitive techniques are required for its detection [11]. Currently, different techniques are available to characterize cfDNA, focusing on either genetic or epigenetic alterations. Profiling of cfDNA methylation has not been studied extensively. In RCC, however, it was previously demonstrated to be a feasible technique to distinguish patients from healthy donors, also at an earlier stage [12].

Here, we define an RCC-specific panel of cfDNA methylation markers using the cfDNA methylation profiles described by Nuzzo et al. [12] in conjunction with the already described RCC methylation markers from the literature [13,14,15,16,17,18]. Subsequently, this panel is evaluated in the serum of patients with mRCC who started WW in the context of a prospective study [7], with the aim of exploring whether cfDNA methylation profiling can be used to identify patients at high risk of rapid disease progression.

## 2. Materials and Methods

### 2.1. Patient Population

All of the patients were recruited within the IMPACT-RCC study (NCT02228954), which was a prospective multicenter cohort study that included patients with histologically or cytologically proven RCC with a clear cell component, as well as recently (<6 months) diagnosed metastases and a good or intermediate prognosis according to the IMDC criteria. At inclusion patients had not received any prior systemic therapy. Patients with untreated central nervous system metastases, patients with symptomatic intra-cerebral metastases and pregnant or breastfeeding women were excluded. As for patients eligible for this study, there was no direct indication to start treatment, the IMDC criterion ‘time from diagnosis to treatment <1 year’ was adapted into ‘time from primary diagnosis to diagnosis of metastatic disease <1 year’. WW was terminated if clinical and/or radiological disease progression was established, in combination with a clinical need to start systemic treatment. However, similarly to the current clinical practice, the treating physician could also decide to continue WW beyond radiological progression if it was found to be clinically appropriate.

### 2.2. Sample Collection and DNA Isolation

Blood was collected in a 10 mL clot activator tube (CAT, BD Vacunatainer^®^, Franklin Lakes, NJ, USA). Within four hours after collection, the blood was centrifuged for 15 min at 2000× *g*, and the serum was stored at −80 degrees until further processing. cfDNA was isolated from the total amount of serum, ranging between 1100 and 4600 µL, using the Maxwell^®^ (MX) RSC ccfDNA Plasma Kit (Promega, Madison, WI, USA). In addition, the serum from 10 age-matched (5 female, 5 male) healthy blood donors (HBDs) was collected and processed following the same protocol. Although the distribution of males and females was not equal between the HBDs and the patient samples, this difference was not statistically different (Fishers exact *p* = 0.1105).

### 2.3. MeD-seq Analysis

For the methylation analysis, 10 ng of cfDNA from the patients was analyzed using the MeD-seq technique, as described before [19]. This technology relies on a methylation-dependent restriction enzyme, LpnPI, that binds to either methylated or hydroxy-methylated cytosines in the following contexts—GmCGC, CmCG, and mCGG—and subsequently cuts the DNA 16 basepairs upstream to generate 32 base pair fragments for sequencing. LpnPI is blocked by a fragment size smaller than 32 base pairs. This unique property prevents the over-digestion of methylation-dense DNA, observed with many other methylation-dependent restriction enzymes and, therefore, allows the accurate analysis of CpG methylation at a single nucleotide resolution. Libraries were multiplexed and sequenced on an Illumina HiSeq 2500 for 50 base pair single reads according to the manufacturer’s instructions (Illumina, San Diego, CA, USA). The samples were first sequenced until ~2 M reads and continued to a total of ~20 M reads, only when the fraction of reads that passed the LpnPI filter (explained below) was at least 20%.

### 2.4. Data Processing

Nuzzo et al. previously described cell-free methylated DNA immunoprecipitation (cfMEDIP) analysis in RCC patients [12]. The authors kindly provided us with the raw sequencing data of 32 stage IV RCC patients and 32 HBDs, which we summarized into 2 kilobase (kb) regions surrounding all known transcription start sites (TSS) annotated in ENSEMBL.

MeD-seq data processing was carried out as previously described [19,20]. In short, the dual indexed samples were demultiplexed using the bcl2fastq software (version 2.20, Illumina, San Diego, CA, USA). Subsequent data processing was carried out using specifically created scripts in Python (version 3.9.11), which include a trimming step to remove the Illumina adapters and a filtering step based on LpnPI restriction site occurrence between 13 and 17 bp from the 5′- or 3′ end of the read, after which only reads originating from the LpnPI digestion remain [19]. The reads passing the filter were mapped to the genome (hg38) using Bowtie 2 (version 2.1.) [21]. Using all of the unambiguously mapped reads, count scores were assigned to each individual LpnPI site in the genome. Subsequently, the count scores for individual CpG sites were summarized into the same 2 kb regions to facilitate comparative analyses.

Only regions containing data in at least 75% of all samples were included in the analysis, resulting in a total of 36,474 regions on chromosome 1–22 for the Nuzzo data, which were all also detected in at least 75% of the IMPACT cohort. The data were normalized to the total number of reads passing the LpnPI filter per sample, after which square root transformation was applied to reduce the skewness in the data distribution. The sequencing data of the Nuzzo dataset is available upon request from the authors, whereas the MeD-seq data from the IMPACT cohort is available in the NCBI Sequence Read Archive (SRA; https://www.ncbi.nlm.nih.gov/sra, accessed on 1 January 2023) under Accession code PRJNA895206.

### 2.5. RCC-Specific Methylation Score

To identify differentially methylated regions (DMRs) between patients and HBDs in the dataset from Nuzzo et al., we used LIMMA (version 3.40.6) [22]. A false discovery rate (FDR) was calculated to correct for multiple testing. The resulting DMRs were intersected with a composed list of 53 potential RCC methylation markers from the literature [13,14,15,16,17,18], resulting in a panel of 22 validated RCC-specific cfDNA methylation markers (Figure 1).

To generate an overall methylation score per patient based on these 22 markers, Z-scores were calculated per region for every patient relative to our normal control panel of 10 HBDs. These Z-scores per region were squared and summed into an RCC-specific methylation score, as described before [20,23] and as explained in more details in our Appendix A. In addition, a detailed description of this method, including the equations, have been added as Appendix A.

### 2.6. Statistical Analysis

The primary aim was to assess the predictive value of a cfDNA methylation score to predict the time to disease progression warranting systemic treatment, also referred to as the WW time, in patients with good or intermediate prognosis RCC. The WW time was defined as the time from the baseline CT to the initiation of treatment. The time to the RECIST-defined progression of disease, also referred to as progression-free survival (PFS), was defined as the time from baseline CT to radiological progression according to RECIST 1.1 [21]. The median follow-up of patients who were still on watchful waiting was 36.2 months (range: 15.8–48 months).

Descriptive statistics were calculated for the variables of interest. A Mann–Whitney U test was performed for the univariate analyses of the continuous variables and a chi-square test was used for the categorical variables. To analyze the association between the methylation score and WW time or PFS, univariate cox proportional hazards analyses were performed using the survival package in R (v3.6.3). To visualize the effect of the significant variables on the WW time and PFS, Kaplan Meier plots were made using the survminer package in R (v3.6.3).

## 3. Results

### 3.1. Building an RCC-Specific cfDNA Methylation Score

When comparing the cfDNA methylation profiles from 31 patients to 31 controls in the Nuzzo dataset [12], we identified 16,713 DMRs with an FDR < 0.1. From these DMRs, we selected the known tissue RCC markers to come to a validated RCC-specific methylation marker panel that was also validated in cfDNA, rather than tissue alone. For this purpose, we composed a list of 53 potential RCC methylation markers from the literature [13,14,15,16,17,18] and checked which of these markers showed significantly differential methylation in the cfDNA dataset obtained from Nuzzo et al. This resulted in an RCC specific, cfDNA specific methylation panel, including 22 RCC markers (Figure 1), which was subsequently summarized into one RCC-specific methylation score.

### 3.2. The RCC-Specific Methylation Score Associates with Clinical Outcome

To evaluate the potential clinical value of this RCC-specific cfDNA methylation score, we turned to the IMPACT-RCC study, which included 42 mRCC patients eligible for watchful waiting (WW). Baseline serum was available in 34 of 42 (83%) patients and the clinical characteristics of the patients from whom baseline serum was available did not differ from the total cohort of patients included in the IMPACT-RCC study (Table 1, Appendix A). The majority of the patients had a pure clear cell carcinoma (79%) and an intermediate prognosis (71%) based on the presence of one risk factor (*n* = 11) or two risk factors (*n* = 13). The median obtained cfDNA concentration was 4.8 ng/mL serum (range: 0.6–117 ng/mL). The MeD-seq analysis was successful in 33 out of 34 patients and all 10 HBDs. The resulting data for the selected 22 markers is shown in Figure 1 and was used to calculate the RCC-specific methylation score in these 33 patients, as well as in the 10 HBDs.

Using the upper limit of the 95% confidence interval observed in our set of HBDs as a cut off for methylation positivity, five patients showed a positive RCC-specific methylation score (Figure 2). From the patients who experienced rapid progression, defined as progression within two months after inclusion (*n* = 4), one had an altered RCC-specific methylation score. However, as shown in Figure 3, we observed that the patients with a score above the cut off had a significantly shorter PFS time (longrank test *p* = 0.0088). For the WW time, the same trend was observed, but the difference in WW time was not significant.

To investigate the potential value of our RCC-specific methylation score in addition to the known prognostic clinical parameters in RCC patients, we performed Cox regression analyses with either the WW time or PFS as the outcome measure. For the WW time, we observed that, in the univariate analyses, only the IMDC score was associated with outcome, whereas age, the number of affected organ sites, the sum of CT-measured diameter of lesions, and our methylation score were not (Table 2a). For PFS, only the methylation score showed a significant association with the outcome in the univariate analyses (Table 2b). As, for both outcomes, only one variable was significantly associated, no multivariable analyses were performed.

## 4. Discussion

The aim of this study was to investigate whether cfDNA could be used as noninvasive biomarker to identify patients with mccRCC with a high risk of early disease progression during a WW period. Therefore, we first established an RCC-specific panel consisting of 22 methylation markers and subsequently evaluated the methylation score for this panel in 33 mRCC patients eligible for WW, based on a low or intermediate prognosis according to the IMDC score. We used a technique that applies a restriction enzyme to generate ≥32 bp methylated fragments for sequencing (MeD-seq). This method was previously shown to enable cfDNA methylation profiling with small amounts of fragmented cfDNA at relatively low costs [19,20].

Although MeD-seq provides a promising method for cfDNA methylation profiling and was found to generate reliable profiles in metastatic colorectal cancer patients, it should be noted the enzyme recognizes ~50% of human CpGs and does not discriminate between methylated cytosines and hydroxy-methylated cytosines. In addition, only methylated DNA fragments are sequenced. Particularly with cfDNA, which is highly fragmentated, the absence of a region could therefore indicate either that the fragment was not present in the sample or that it was not methylated.

We found that an altered methylation score, relative to the HBDs, was significantly associated with a shorter PFS, but not associated with the WW time. The results from this relatively small cohort of patients suggests that the blood-based methylation scores might add to the IMDC score for the identification of patients with low- and intermediate-risk mRCC who will show rapid disease progression.

The subset of mccRCC patients managed with WW is not often described in the literature. While WW is an acknowledged treatment strategy in the current international guidelines, there are also no well-validated criteria to identify mccRCC patients for this strategy [5]. The time from the initial RCC diagnosis to the development of metastatic disease, a factor considered to be an indicator of poor prognosis in patients on systemic treatment, was not prognostic in patients on WW [24]. In all prospective studies on WW, patients with a good or intermediate prognosis according to the IMDC score could be considered for a WW-strategy. The large diversity in the duration of WW underlines that the IMDC criteria alone is unable to distinguish between patients with an expected long or short period of WW [6,7,24]. It is encouraging that two studies have shown that the identification of patients eligible for WW can be improved with the knowledge on the disease extent at diagnosis (number of involved organs). However, the number of patients in these trials limit the impact of these findings. Exploring other non-invasive biomarkers to distinguish indolent from more aggressive disease is therefore desired.

As such, several other researchers have investigated the prognostic value of tumor signal presence in cfDNA, also referred to as circulating tumor DNA (ctDNA), in patients with RCC. There is some evidence suggesting that the ctDNA level prior to surgery in the earlier stages of disease is associated with higher recurrence rates [25]. Alternatively, the presence of ctDNA has also been associated with an inferior outcome in the treatment of metastatic disease [26]. These were, however, only small cohorts of patients, and solid data on the prognostic value of ctDNA presence is still lacking. In addition, various methods have been used for the detection of ctDNA, varying from somatic mutations to untargeted methods such as copy number alterations [10]. As renal cell carcinoma is known to be characterized mainly by epigenetic alterations, methylation profiling therefore poses a sensible approach for the detection of tumor DNA in blood [12,27]. Therefore, more recent work has focused on methylation approaches for ctDNA detection in RCC patients. The advantages of methylation analysis have been demonstrated in more recent work showing that methylation profiling identified the presence of ctDNA at the earlier stages of disease [12].

For our study, it is remarkable that only fifteen percent of the patients in our study that included only patients with stage IV disease, were methylation positive using our approach. This might be explained by the fact that mccRCC patients were previously found to have low ctDNA amounts in general [28,29,30]. Moreover, we used serum and not plasma as a source for our cfDNA analyses for availability reasons, which likely complicated the detection of tumor specific methylation patterns, as serum is known to contain lower tumor fractions due to a higher contamination of DNA originating from leukocytes [31]. Other studies investigating cfDNA in stage IV disease found larger proportions of patients to have detectable ctDNA, ranging between around 30% [10,32] and 100% [12]. The fact that we used serum and not plasma is a limitation of our study and was related to the availability of materials. In addition, the sample size of our study is relatively small, although the majority of patients had serum samples available. The lack of a large series of RCC in patients in the literature probably reflects the challenge of performing clinical studies in patients with RCC. To our knowledge, however, this is the first series that has related cfDNA methylation patterns to clinical outcome in mccRCC patients.

Moreover, it is important to note that we found the methylation score to be significantly associated with the PFS, but not with the WW time. As per the protocol, some patients who progressed according to RECIST 1.1 did not start with systemic treatment directly, explaining the difference between the observed PFS and WW times. The main reasons not to start treatment were the progression of overall small sized metastatic lesions or lesions growth not leading to overt symptoms; the decision not to start treatment in these cases reflects the current daily practice. This raises the question of whether a biomarker that is better correlated with radiological progression than with WW time will ever be clinically relevant in this setting, even if it outperforms the IMDC score.

## 5. Conclusions

Our study demonstrates that RCC-specific DNA methylation is detectable in the blood of part of the patients and appears to be associated with disease progression. Together, these results provide a proof-of-concept that cfDNA methylation analysis could be used as a prognostic tool in RCC patients. For the WW time, we only observed a non-significant trend towards an association with the methylation score. Although this study did not find definite proof that cfDNA methylation patterns are associated with WW time, this should be validated in larger studies, in which plasma should be collected prospectively. Furthermore, its additional value to the IMDC score remains to be elucidated.

## Figures and Tables

**Figure 1 cancers-15-01374-f001:**
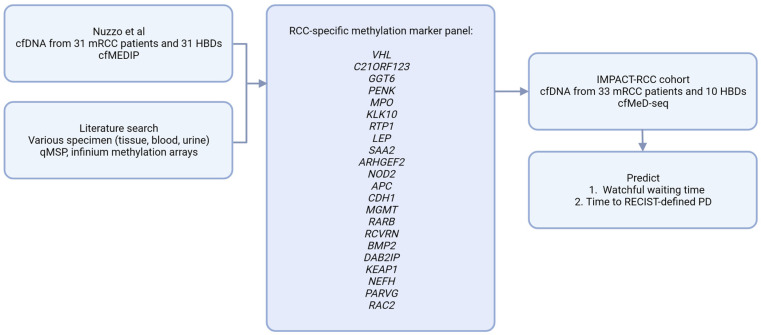
Analytical approach. cfMEDIP; Cell-free methylated DNA immunoprecipitation, cfMeD-seq; Methylated DNA sequencing, HBD; Healthy blood donor, PD; progressive disease, RCC; Renal cell carcinoma, qMSP; quantitative methylation specific PCR.

**Figure 2 cancers-15-01374-f002:**
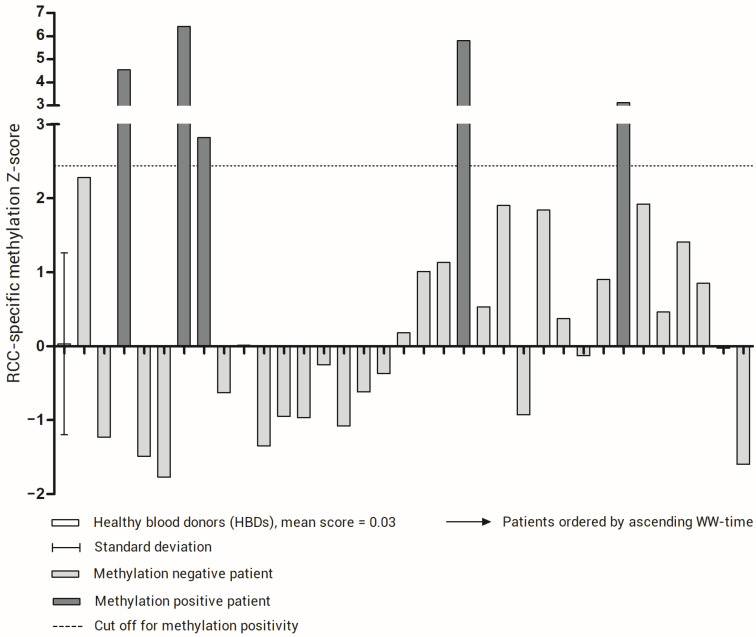
RCC-specific methylation score for healthy blood donors (HBDs) and patients. For HBDs, the mean and standard deviation is indicated. The cut-off for methylation positivity, represented by the upper limit of the 95% confidence interval of the methylation score in HBDs, is indicated by a dotted line. The patients are ordered by WW time.

**Figure 3 cancers-15-01374-f003:**
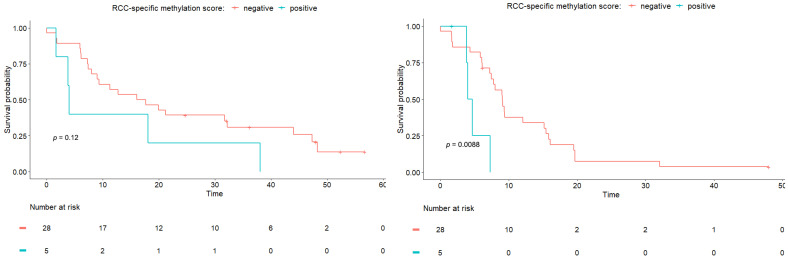
Kaplan Meier analyses for WW time (**left**) and PFS time (**right**). Patients are divided based on RCC-specific methylation score. A log-rank test was performed to compare the curves.

**Table 1 cancers-15-01374-t001:** Patient demographics and clinical characteristics.

Characteristics	Patients in IMPACT-RCC Study (*n* = 42)	Patients with Serum Samples Successfully Analyzed (*n* = 33)	*p*-Value
Gender			
Male, *n* (%)	31 (74%)	26 (76%)	
Female, *n* (%)	11 (26%)	7 (21%)	0.786
Median age in years (range)	66 (44–86)	66 (44–86)	0.790
Nephrectomy			
Yes, *n* (%)	36 (86%)	28 (85%)	
No, *n* (%)	6 (14%)	5 (15%)	1
Histology			
Pure clear cell, *n* (%)	32 (76%)	27 (79%)	
Mixed, *n* (%)	10 (24%)	6 (18%)	0.586
Time from diagnosis to first metastasis			
<1 year, *n* (%)	23 (55%)	19 (58%)	
≥1 year, *n* (%)	19 (45%)	14 (42%)	0.820
IMDC risk factors			
0 (favorable)	14 (33%)	10 (30%)	
1 (intermediate)	13 (31%)	10 (30%)	
2 (intermediate)	15 (36%)	13 (39%)	0.941

**Table 2 cancers-15-01374-t002:** Cox regression analysis. The tables were adopted from the main paper and updated according to the patients included within the biomarker analysis. The methylation scores resulting from the biomarker analysis were added as variables.

	Variables	HR	95% CI HR	*p*-Value
(a) Cox proportional hazards regression analysis results for watchful waiting time.				
Clinical	Age	1.00	0.96–1.024	0.99
	IMDC criteria, per unit increase in score	2.01	1.19–3.41	0.01
	Number of affected organ sites per patient	1.43	0.96–2.11	0.07
	Sum of CT measured diameter of lesions per patient (cm)	1.02	1.00–1.05	0.08
Molecular	RCC-specific methylation score	2.18	0.81–5.87	0.12
(b). Cox proportional hazards regression analysis results for time until RECIST-defined PD				
Clinical	Age	1.01	0.96–1.05	0.82
	IMDC criteria, per unit increase in score	1.35	0.86–2.10	0.19
	Number of affected organ sites per patient	1.07	0.73–1.57	0.74
	Sum of CT measured diameter of lesions per patient (cm)	1.01	0.99–1.04	0.30
Molecular	RCC-specific methylation score	4.45	1.32–15.05	0.02

HR; Hazard ratio, CI; Confidence interval, IMDC; International Metastatic Renal Cell Carcinoma Database Consortium, CT; computed tomography.

## Data Availability

The sequencing data are available in the NCBI Sequence Read Archive (SRA; https://www.ncbi.nlm.nih.gov/sra accessed on 15 February 2023) under Accession code PRJNA895206. The clinical metadata are available from the corresponding author upon request. The sequencing data of 32 stage IV RCC patients and 32 HBDs described by Nuzzo et al. [9] were requested from the corresponding author.

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
