# Peer review of "Methylated Cell-Free DNA Sequencing (MeD-seq) of LpnPI Digested Fragments to Identify Early Progression in Metastatic Renal Cell Carcinoma Patients on Watchful Waiting"

_cancers, 2023, doi:10.3390/cancers15051374_

Round 1

Reviewer 1 Report

Thank you for the opportunity to review the manuscript entitled “Methylated cell-free DNA sequencing to identify early progression in metastatic renal cell carcinoma patients on watchful waiting” The authors investigate a panel of 22 RCC-specific methylation marker markers  for association with rapid progression in renal cancer patients. This is an interesting topic where a lot of information is available. In general, the article is well organized and structured. However, I would like to encourage minor modifications. The results should be validate on a larger number of patients.  Are the authors planning to test the method on a larger cohort?

Materials and Methods

•           Statistical part needs to be revised. Line 196 Kaplan-Meier curves ??

 Results

Table 1. Please correct N or n

 Figure 2. Please indicate the methylation score for the healthy blood donners. Adjust the y-axis with gaps and segments

Line 272:  The right number of samples analyzed was 33.  Could please the authors correct or clarify (Line 215)?

It would be important to comment in the discussion the stating of the Tumor and the positive RCC-specific methylation score (n=5).

 I suggest including a limitation part in the article

Author Response

We thank the reviewer for reviewing this manuscript and providing us with suggestions. We carefully adressed them, please find the responses attached. 

Reviewer 2 Report

The author's purpose to use the methylated cell-free DNA sequencing data as a biomarker of early progression in metastatic-RCC patients. The work is very interesting and tries to provide new insights for better patients stratification. However, some points need to be clarified:

1. In the introduction the authors should explore the differences in terms of cfDNA of different fluids, i.e. plasma vs serum.

2.  In Patient Population  should be clear the number of patients included and the characteristics of HBD (age, gender,..). Additionally in patients group needs to be clear how many patients had rapid progression and the critierias for that classification.  

3. The equation used for RCC-Specific Methylation Score calculation must be included.

4. According to authors, HBD group is composed by 5 Male and 5 females. However the patients groups presents a different distribution. This could influence the results, so the authors need to assess if the gender is not statistical different between patients and HBD. 

5.  In figure 2 would be useful for the readers if the authors identify the patients that progress.

6. In the manuscript the authors only performed a univariate analysis, a multivariate analysis should be applied.

7. the discussion section needs to be improved and correlate with similar data used in other pathologies, and to explore the clinical applicability and relevance of paper's findings.

Author Response

We thank the reviewer for reviewing this manuscript and provinding us with suggestions to improve it. We have carefully adressed the suggestions, please find our response attached. 

Reviewer 3 Report

The authors have utilised a low resolution methylation analysis (MeD-seq) to characterise patients on watchful waiting diagnosed with renal-cell carcinoma. The aim was to investigate the likelihood for methylation information to be informative to detect early the patients who are going to progress quicker with their disease.

Main comments

Overall I have no concerns about the writing style of the manuscript and background, also the presentation of data and plots. My main concern is about the chosen assay versus the expectations towards such a low resolution enrichment-based method. This technique is quite poor in performance in comparison to base-resolution techniques, and this is not reflected anywhere in the MS, whilst it is so key that I would deem it important to include in the title, which currently gives the wrong impression that a base-resolution sequencing is used. More appropriate is “Methylated Cell-Free DNA Enrichment Sequencing”. My point is, the authors look for promoter changes (select 2kb regions around the TSS), and those can be quite variable in cancer – the technique utilised would only have the potential to capture very drastic changes, and this seems to be the case (quick progression for all). Restriction-based techniques have very high background noise, and this noise may easily mask the real results, which are often too variable and inconsistent in cancer anyway. Moreover, the technique used does not discriminate between hmC and mC, and for promoter regions this is an important distinction. hmC is already very low in cancer so interference with the final signal calling may be low here, but nevertheless it is important to discuss – lower risk patients may retain more hmC since hmC overall is a better early cancer prognostic marker than mC. And here this would be confused with mC and will result in higher score. Whilst the paper referred to for biomarker validation only methylation enrichment technique was used (meDIP-seq – no hmC enrichment there). This needs to be clarified in discussion. Also cf DNA is very fragmented already and further enzymatic fragmentation inevitably leads to high loss of material. All these downsides should be explained in the discussion.

And I actually believe higher-resolution approaches would have given different, possibly more informative, results. For future testing with an economic technique I would recommend looking into amplicon-based assays at base-resolution (bisulfite or enzymatic treatment) - multiplexed or single-plex – also much more appropriate for low content cfDNA, no further fragmenting and loss, there are papers online to look for. Serum vs plasma is no concern to me, there is enough cfDNA there for the right technique. 

I’d also favour giving a brief explanation about data processing and mapping, and mC calling, despite cited references.

Minor comment:

Figure 1 legend – “HBD; Healthy blood donor” repeated twice

Author Response

(The authors gave the same response as above.)

Round 2

Reviewer 2 Report

The authors answered the raised concerns.